# Natural Rubber Blend Optimization via Data-Driven Modeling: The Implementation for Reverse Engineering

**DOI:** 10.3390/polym14112262

**Published:** 2022-05-31

**Authors:** Allen Jonathan Román, Shiyi Qin, Julio C. Rodríguez, Leonardo D. González, Victor M. Zavala, Tim A. Osswald

**Affiliations:** 1Polymer Engineering Center, Department of Mechanical Engineering, University of Wisconsin-Madison, Madison, WI 53706, USA; tosswald@wisc.edu; 2Scalable Systems Laboratory, Department of Chemical and Biological Engineering, University of Wisconsin-Madison, Madison, WI 53706, USA; sqin34@wisc.edu (S.Q.); gonzlezchvez@wisc.edu (L.D.G.); victor.zavala@wisc.edu (V.M.Z.); 3SOAN Laboratories, Bogotá, Colombia; jcero@soanlaboratorios.com

**Keywords:** viscoelasticity, machine learning, response surface methodology, natural rubber, reverse engineering, formulation, optimization, modeling

## Abstract

Natural rubber formulation methodologies implemented within industry primarily implicate a high dependence on the formulator’s experience as it involves an educated guess-and-check process. The formulator must leverage their experience to ensure that the number of iterations to the final blend composition is minimized. The study presented in this paper includes the implementation of blend formulation methodology that targets material properties relevant to the application in which the product will be used by incorporating predictive models, including linear regression, response surface method (RSM), artificial neural networks (ANNs), and Gaussian process regression (GPR). Training of such models requires data, which is equal to financial resources in industry. To ensure minimum experimental effort, the dataset is kept small, and the model complexity is kept simple, and as a proof of concept, the predictive models are used to reverse engineer a current material used in the footwear industry based on target viscoelastic properties (relaxation behavior, tanδ, and hardness), which all depend on the amount of crosslinker, plasticizer, and the quantity of voids used to create the lightweight high-performance material. RSM, ANN, and GPR result in prediction accuracy of 90%, 97%, and 100%, respectively. It is evident that the testing accuracy increases with algorithm complexity; therefore, these methodologies provide a wide range of tools capable of predicting compound formulation based on specified target properties, and with a wide range of complexity.

## 1. Introduction

The use of naturally derived or sustainable polymers is continuing to gain increasing attention due to the long-term environmental harm that their synthetic analog imposes. These synthetic polymers over the last few decades have become the most practical and economical solution for a huge variety of applications across multiple industries given their cost savings [1,2,3]. Such ease of processing, synthetic polymers and having more readily modifiable properties make them a far more attractive choice for industries looking to maximize profit and production efficiency, and not taking into priority the environmental effects of this decision. Consequently, the current understanding of biodegradable and biopolymers lags behind synthetic polymers. Furthermore, the ease of applicability of synthetic polymers within the business model comes at the expense of the environment as synthetic polymers are derived from a non-renewable resource, resulting in an increased carbon footprint [4,5,6]. With 27 countries implementing a carbon tax such as México, Colombia, the EU, and China to name a few, industry will be encouraged to embrace polymers that create a negative carbon footprint. An example of such a biopolymer is natural rubber, which is derived from the *Heveas brasilensis* tree as latex, polyisoprene which is commonly made up of about 59% water and 36% rubber solids [7,8,9,10].

With the appropriate additives, natural rubber can be manipulated to acquire distinct properties for specific applications. Previous studies have shown how increasing sulfur content increases crosslink density and hardness, leading to a less viscoelastic material [11,12,13]. From these studies, Zhao and coworkers were able to demonstrate that the Shore A and 300% modulus of NR vulcanizates both increased linearly with crosslink density, but dynamic properties still vary in a non-linear fashion [11]. Plasticizers are another commonly used additive in industry where it is known to decrease viscosity and improve processability but also decrease tensile properties, tear strength, resilience, and compression set [14,15,16]. Furthermore, sodium bicarbonate is an additive commonly used for the light-weighting of parts as the elevated temperatures during vulcanization trigger degradation of the additive, resulting in the release of carbon dioxide. The release of gasses within the blends creates the foaming effect where a porous inner structure is produced [17,18]. It is important to recognize that introducing a foamed structure to a testing specimen under compression behaves very differently from a foamed testing specimen under tension. The act of compressing testing a foamed polymer structure results in a typical compression curve seen in Figure 1 below where there are three zones, the initial zone with a higher tangent modulus, the elastic buckling zone, and the densification zone. The densification region is where the cellular structure begins to collapse on itself, and the stresses begin to grow as gas and polymer are compressed. 

Furthermore, physical or chemical blowing agents may bring about variation depending on the cell size and the number of cells present within the part [19,20,21]. Both morphological properties create variability in material properties, therefore, it is crucial to understand the level of influence that foaming partakes in mechanical behavior. It is also important to note that all additives have interacting effects and that varying chemical grades also bring about variation to your blend material properties. The sensitivity of material properties due to additives is a complex issue that deems formulating blends a highly complex task. 

Currently, industry leans on the expertise of a formulator and highly depends on that individual to arrive to the final formulation with the least number of iterations to the blend by trial and error. This study focuses on using machine learning algorithms and statistical methods to predict and determine the optimal blends based on target properties that are indicative of long-term and short-term mechanical behavior. The implementation of machine learning within polymer processing has reached various areas within additive manufacturing in regard to extrusion behavior [22,23], prediction of injection molding events [24,25,26], and post-production material behavior [27,28,29,30]. The common denominator for above-mentioned applications is that they require real experimental data to aid in the creation of representative models. With polymers exhibiting time- and temperature-dependent behavior, it is crucial to quantify the long-term and short-term behavior of the polymer of interest to fully understand its material behavior. For that reason, viscoelastic properties (relaxation behavior and tanδ), hardness, and the quantity of voids in the sample were selected as the measured properties as they characterize the morphology, short-term behavior, and long-term behavior of polymers [31,32,33].

Relaxation measurements allow for long-term behavior analysis as it can also be interrelated to creep behavior [33,34,35]. Moreover, dynamic properties are crucial for understanding mechanical behavior for high-performance applications undergoing cyclical loading [36,37,38,39]. For example, footwear materials undergo cyclical loading between 1 Hz –5 Hz while materials within the automotive industry may experience cyclical loading within a frequency range of about 0 and 300 Hz [40]. Furthermore, it should be noted that with the automotive industry embracing the electrification of automobiles, it should be expected to encounter higher-frequency vibrations within the automobile, supporting the need to understand the frequency-dependent behavior of polymers implemented in automotive design [40]. Additionally, with foaming technologies entering high-performance applications due to their role in light-weighting, it is no surprise that the addition of air bubbles within a polymer matrix will alter material performance [18,41,42]. 

The above-mentioned material characteristics deem formulating a highly complex process as one additive may increase one material property but non-linearly decrease another. It is a process that requires the formulator to balance the interacting effects of all additives, while ensuring that the cost of the overall blend is minimized without sacrificing quality. This study will focus on providing industry with three data-driven formulation methodologies, varying in complexity (Response Surface Method, Artificial Neural Networks, and Gaussian Process Regression) to explain how certain algorithms can offer additional information to the process besides treating the algorithm like a black box. The final goal of this study is to use the algorithms as a method for reverse engineering current materials based on their viscoelastic properties (relaxation, dynamic, and hardness). The reverse engineering aspect of this study is crucial for formulators as they can feed the algorithm a starting point to their design of experiments. 

## 2. Materials and Methods

### 2.1. Materials and Blending

SOAN Laboratories provided the Polymer Engineering Center with Betapreno, one of their ammonia-free natural rubber (NR) materials produced in Victoria, Colombia. Table 1 below depicts the additives used within this study to understand the influence of commonly used additives on the mechanical behavior of natural rubber blends, primarily its viscoelastic properties. Blends were prepared in a C.W. Brabender 3-Piece bowl mixer using Banbury blades with a chamber volume of 75 ${\mathrm{cm}}^{3}$. Table 2 below shows each blend prepared within this study, where the last two blends were outputted by the predictive algorithms as target blends, used to improve the prediction accuracy of the machine learning algorithm. All blends were mixed at 21 °C and 50 RPMs until a homogeneous mixture resulted in the plateau of the torque response logged by the Intelli Plasti-Corder Torque Rheometer mixer attachment. Additives were tested at varying load ranges to capture the effect of a low load amount to a large load amount. This was done to ensure the algorithm captures the limiting effect of an additive as properties do not always trend in the same manner as you increase the amount of additive [43,44,45,46]. Jacob and coworkers showed that increasing reinforcement increased the tensile strength of the natural rubber blend only if the loading was below 30 pph (parts per hundred of rubber) [47]. After exceeding 30 pph of fiber loading, the tensile strength would begin to decrease with fiber loading. Capturing this phenomenon is crucial as the algorithm needs to understand when loading becomes detrimental to certain properties. It is also important to note that only a small number of blends were created to prove that a small DOE (design of experiments), given the right data, will be able to aid in the creation of an accurate predictive algorithm. 

### 2.2. Material Characterization

#### 2.2.1. Rheological Characterization

A TA Instruments AR-2000EX parallel plate rheometer was employed to quantify the vulcanization reaction of each natural rubber blend. This instrument has a displacement resolution of ±40 nrad and a torque resolution ±1 nNm. By using the ASTM D2084 standard, the reaction was characterized to determine the optimal vulcanization time, $t_{90}$, at 150 °C. Furthermore, to prevent slippage-induced measurement error, parallel plate rheometry was conducted using serrated parallel plates. After rheological testing, the samples were placed in cylindrical molds and vulcanized in the Carver 3889 compression molding machine by exerting 1000 lbs of force at 150 °C at the specified $t_{90}$.

#### 2.2.2. Lissajous Curve Characterization

The NETZSCH Eplexor^®^ 500 N Dynamic Mechanical Analyzer (DMA) was utilized for the characterization of Lissajous curves and tan δ for each specific sample. NETZSCH’s DMA is capable of both transient and dynamic testing within a frequency range of 0.01 Hz and 100 Hz. For this dynamic study, a 10% static compressive strain was imposed on a cylindrical sample with 10 mm diameter and 10 mm height, and a ±5% dynamic strain was applied at 1 Hz. The large levels of strains classify these testing conditions as a large amplitude oscillatory test (LAOS) which creates a valid representation of what stresses and strains are experienced during walking. The shape of the Lissajous curve gives us an insight into the level of non-linearity present when loaded cyclically, and also gives us the tanδ (Figure 2), a measurement represented by the lag between the stress and strain response during testing, correspondingly the ratio between the Loss Modulus and Storage Modulus (Equation (1)).
(1)$$\tan\delta =\frac{E^{\prime \prime }}{E^{\prime }}=\frac{\mathrm{Loss}\;\mathrm{Modulus}}{\mathrm{Storage}\;\mathrm{Modulus}}$$

#### 2.2.3. Relaxation Curve Characterization

Similarly, to dynamic testing, relaxation behavior was characterized with the NETZSCH Eplexor^®^ 500 N DMA by imposing a 30% compressive static strain for 10 min and logging the stress response from the material with respect to time. Strain of 30% was selected as the testing condition since 10% resulted in the same normalized curve, where a large signal-to-noise ratio is present, as seen in Figure 3. As the material is quickly loaded in compression to the predefined strain, the stress reaches the maximum point and begins to decay with time. 

The relaxation time is commonly defined as the time necessary for the material to reach 1% of the maximum stress reached when the static strain was applied at the start of [31,48,49]. Furthermore, given that full relaxation of a cross-linked polymer is not frequently attainable at time scales below 10 h, as seen in Figure 4 below, characterization of the relaxation curve was done by tabulating the maximum stress during relaxation testing and by fitting the decay of the curve with a power function. Most polymers during relaxation can be modeled with a power function (Equation (2)) where $n_{\mathrm{relax}}$, is used to quantify the rate at which stress decays [33]. By analyzing the magnitude of $n_{\mathrm{relax}}$, it can be deduced that if a $\left|n_{\mathrm{relax}}\right|$ is large, then the decay occurs more rapidly compared to a material with a smaller $\left|n_{\mathrm{relax}}\right|$.
(2)$$\sigma_{\mathrm{relax},\mathrm{normalized}}=A\;\cdot \;{\mathrm{time}}^{-n_{\mathrm{relax}}}$$

#### 2.2.4. Sample Morphology Characterization

Before mechanical testing, the ZEISS Metrotom 800 μCT (Carl Zeiss AG, Oberkochen, Germany) was used to scan (1 h) each cylindrical sample and to create a 3-dimensional image of the cylindrical specimen by using the scanning parameters mentioned in Table 3.

Each scanned image represents a cross-section of the sample with pixel intensity values ranging from 0 to 255. A pixel intensity value of 0 corresponds to the color black while a pixel intensity value of 255 corresponds to the color white. A material with a higher density will have a much higher pixel intensity value compared to a void, which will output a lower pixel intensity value. For that reason, a MATLAB program, as schematically shown below in Figure 5, was developed to threshold each image and to characterize the quantity of voids within each cross-section, allowing for a through-thickness analysis of each cylindrical sample. 

#### 2.2.5. Durometer Characterization 

A durometer (Shore A) was used to characterize the hardness of each respective blend. It is important to take into consideration that the hardness measurement depends greatly upon the indenter geometry and the amount of force applied by the operator. In accordance with ISO Standard D2240-15, this Type A indenter shape includes a flat cone point (0.79 mm), and a 35° included angle with a range of 0~100 HA.

### 2.3. Computational Methods for Predictive Models

#### 2.3.1. Linear Regression Implementation

Multivariate linear regression is one of the simplest models to construct a mathematical expression of the material property as a function of blend content. Each of the target properties—durometer reading, $\sigma_{\mathrm{relax}}$, $n_{\mathrm{relax}}$, and tanδ—is formulated by Equation (3):(3)$$Y=a_{0}+a_{1}A+a_{2}B+a_{3}C$$

In the model above, A corresponds to sulfur content, B to paraffin oil content, and C to the void content within the sample; the coefficients (or weights and biases) $a_{i}$’s are determined by ordinary least squares [50]. The model was implemented using Scikit-learn [51] (version 0.24.2) with Python.

#### 2.3.2. Response Surface Method Implementation

Instead of linear regression, the response surface method (RSM) uncovers the interconnectivity between various controllable factors and several response variables using nonlinear modeling. It should be acknowledged that RSM serves as an approximation method that provides a relatively easy method for modeling, estimating, and optimizing based on target parameters [52,53,54]. By use of mathematical and statistical techniques, an empirical model is created from experimental data and is used to evaluate the fit to a statistical model (linear, quadratic, cubic or two-factor Interaction), as described in Equation (4) below.
(4)$$Y=a_{0}+\underset{\mathrm{Linear}}{\underbrace{a_{1}A+a_{2}B+a_{3}C}}+\underset{\mathrm{Quadratic}}{\underbrace{a_{4}A^{2}+a_{5}B^{2}+a_{6}C^{2}}}+\underset{\mathrm{Two-Factor}}{\underbrace{a_{7}\mathrm{AB}+a_{8}\mathrm{AC}+a_{9}\mathrm{BC}}}$$

Same as above, the independent variables A, B, and C represent sulfur content, paraffin oil content, and void content, respectively. The coefficients ($a_{0}-a_{6}$) determined by the model within the linear and quadratic sections dictate the influence each respective variable has on the output, Y, while the Two-Factor coefficients ($a_{7}-a_{9}$) of the quadratic model above quantify the level of influence that interactions between two variables have on the output [52,53,54,55]. As mentioned above, formulating is a balance between various additives and RSM allows for the user to determine, to some degree, the interaction effects between two controllable variables. 

Once the model is created, RSM allows for the optimization of a blend based on target responses, maximizing a specific response, or minimizing a specific response. The options available via RSM are of value as some additives may result in a dramatic increase in raw material expenditure, therefore, one can have a combination of maximizing certain controllable factors and minimizing others to ensure cost rises are mitigated. Additionally, each target response may be given an importance value ranging from 0.1 to 10 if the user is more interested in some responses compared to others. If all target responses are of equal importance, then the default value should be 1, and if one target response is crucial for the user, then the importance value can be increased up to a value of 10. Furthermore, weights may be assigned to each target response with values ranging from 0.1 to 10. The weights influence the desirability function shape between the lower/upper bounds and the target. Figure 6 below shows how different weights may influence the desirability function shape where it can be seen that a larger weight creates a shaper, and quicker convergence to the target response. 

Although the quadratic function is mathematical, the statistical analysis of this regression model is key for interpreting the model. Minitab^®^ 20 was used for the RSM study whereas a 95% confidence level was selected as the threshold for statistical significance. The Pareto chart of the Standardized Effects lists the standardized coefficients to understand which terms have the highest influence on each given response. By observing the magnitude of the standardized effect, the user can understand which controllable factors are the most influential and if a combination of controllable factors creates significant change to the output response. The dashed line within the Pareto chart indicates a significance level of α=0.05, therefore, anything to the right of the line suggests a term that is statistically significant to the response [52,53,54,56].

Additionally, the Coded Coefficients, such as the coefficients and *p*-value for each respective controllable factor allow the user to determine if a specific term is significant. If *p*-value ≤α, the association is statistically significant, and if *p*-value >α, the association is not statistically significant. Finally, the Model Summary is described by the standard deviation of the distance between fit values and input data values (S), the percentage of variation within the model response ($R^{2}$), the adjusted $R^{2}$ which is the variation in the response adjusted for the number of predictors in the model relative to the number of observations ($R_{\mathrm{adjusted}}^{2}$), and the $R^{2}$ of the predictions which indicates how well the model predicts the removed observations ($R_{\mathrm{predictions}}^{2}$). It is important to note that if $R_{\mathrm{predictions}}^{2}$ is substantially less than $R^{2}$ then this may indicate that the model tends to overfit. 

#### 2.3.3. Artificial Neural Network Algorithm Development

Artificial neural network (ANN), which has added nonlinearity compared to RSM, is a powerful modeling method that can approximate highly nonlinear functions [50]; it can be used to explore complex relationships between experimental design parameters and material properties. The basic architecture of a fully connected ANN can be seen in Figure 7, and consists of an input layer, hidden layer(s), and an output layer. The input layer takes the design parameters $x={\left[A\;B\;C\right]}^{T}$, including the sulfur, paraffin oil, and void content, which is then mapped to a high-dimensional space in each hidden layer using linear transformation followed by nonlinear activation represented as $y_{h}=\sigma \left(W^{T}x+b\right)$. At the end of the architecture, an output layer is incorporated for predictive purposes by summarizing the mapped features from the hidden layers to the desired output y through a linear transformation. During model training, the model parameters (weights W and biases b) of the network are updated iteratively to minimize the difference between the predicted and experimental values (loss function). Backpropagation [50] serves as the automatic differentiation algorithm to calculate the gradients of the loss function with respect to the model parameters, and the gradients are then used to update the model parameters in the loss-reducing direction.

In this study, an ANN architecture was constructed using Tensorflow [57] (version 2.0.0) and tuned for each of the four material properties, including durometer reading, $\sigma_{\mathrm{relax}}$, $n_{\mathrm{relax}}$, and tanδ. To save experimental costs, the dataset for training and validation was kept relatively small, which contains 326 data instances. As a result, the hyperparameters of the ANN models were tuned with several considerations that avoid overtraining, such as the incorporation of dropout layers and early stopping as well as the leverage between the number of model parameters and cross-validation accuracy. To find the optimal set of hyperparameters, a grid search was performed. The grid search considers all the possible combinations of the discretized hyperparameters in the pre-constructed search space—including learning rate (0.001, 0.003, 0.01), number of hidden layers (1, 2, 3), and number of neurons (16, 32, 64, 128, 256)—and evaluates the model performance with each NN architecture. The model parameters were optimized using the mean absolute error and the Adam optimizer [58], a commonly used optimizer in industry. 

Besides the predictive power, the relationship between the blend characteristics and the material performance may be inferred from a trained ANN. This can be achieved through a gradient-based sensitivity analysis that studies the influence of individual blend content on each of the material properties. In the sensitivity analysis, the gradients of the output (each material property) with respect to the input (blend content) are computed and summarized for each input variable. The gradient values can indicate how blend content contributes to certain material performance, which may lead to human-informed design logistics for new materials.

#### 2.3.4. Gaussian Process Regression Algorithm Development

Gaussian process regression (GPR) [59] is another class of supervised learning algorithms. Unlike ANNs, GPR makes probabilistic predictions in the form of mean and variance, and therefore uncertainties of the predictions can be calculated. With such information, one can potentially design additional experiments that target the region with high uncertainty. Another advantage of GPR lies in its “non-parametric” nature, and minimal hyperparameters need to be learned. Compared with ANNs whose number of parameters can easily add up to thousands and millions when increasing layers, GPRs usually carry fewer hyperparameters to tune. In general, GPR loses efficiency for high-dimensional data but works well with small datasets. Mathematically, GPR utilizes a Gaussian process prior that is specified by the means and covariance matrices of the training and validation/testing data. The covariance matrices (hyperparameters) are optimized during the training process and serve as a way to approximate the similarity between a test input and the training inputs, which eventually leads to the prediction of the test output based on the training outputs through matrix operation [59]. 

As illustrated in a one-dimensional example in Figure 8 below, GPR learns from the given data points (training data) and predicts the “y” values with a mean and variance. The region with training data shows a narrower variance than the region without training data. Based on the mean and variance, a sample curve (dashed line) can be drawn from the predicted distribution. 

In this study, the GPR model was implemented using Scikit-learn [51] (version 0.24.2); a basic radial basis function (RBF) kernel was used, and the hyperparameters (length scale parameters) were optimized during data fitting. Again, 5-fold cross-validation was used to evaluate the model performance.

## 3. Results and Discussion

### 3.1. Material Characterization

#### 3.1.1. Influence of Voids on Viscoelastic and Static Properties

The degradation reaction of sodium bicarbonate within the blend has the ability of creating a foamed part with varying levels of void amount depending on the additive load and charge within the compression molding mold [54]. The micrographs seen in Figure 9 below show the three examples of void amounts captured by the ZEISS μCT scanner where the amount of sodium bicarbonate was kept constant and charge weight within the mold was altered. 

It is evident that increasing void amount results in a decrease in solid material carrying load during testing [18,41,42]. For that reason, it is expected to see that a foamed structure consisting of the same formulation will result in a less rigid material, compared to one of lesser voids. Figure 10 below shows the overlay of relaxation tests for the same formulation but at varying void contents and it can be observed that increasing voids results in a vertical shift of the overall curve. Furthermore, in Figure 11 below, a clear linear relationship between the rigidity of the material and void amount can be seen if the maximum stress reached during testing is grouped with its respective void amount. 

Voids have a present influence on how rigid the material is during relaxation, and it can also be observed that the rate at which stress decays is also influenced by the quantity of voids present in the sample. Evaluating the $n_{\mathrm{relax}}$ of individual curves allows for the analysis of how voids influence the rate of stress decay, whereas it can be seen in Figure 12 how voids and $n_{\mathrm{relax}}$ have a linear decreasing relationship. This relationship states that as the amount of voids increases, the rate at which stress decays will be larger, therefore, if a material with minimal compression is sought after then the amount of voids would need to be minimized.

To better understand at a quantitative level the amount of influence that parameters impose on a certain material property, the Pearson correlation coefficient is utilized to determine the level of significance between two specified variables. The correlation matrix is constructed by using the Equation (5) below whereas x may represent sulfur content and y could be voids.
(5)$$r=\frac{\Sigma \left(x_{i}-\bar{x}\right)\left(y_{i}-\bar{y}\right)}{\sqrt{\Sigma {\left(x_{i}-\bar{x}\right)}^{2}\Sigma {\left(y_{i}-\bar{y}\right)}^{2}}}$$
r=correlation coefficient
$$x_{i}=\mathrm{values}\;\mathrm{of}\;\mathrm{the}\;x-\mathrm{variable}\;\mathrm{in}\;a\;\mathrm{sample}$$
$$\bar{x}=\mathrm{mean}\;\mathrm{of}\;\mathrm{the}\;\mathrm{values}\;\mathrm{of}\;\mathrm{the}\;x-\mathrm{variable}$$
$$y_{i}=\mathrm{values}\;\mathrm{of}\;\mathrm{the}\;y-\mathrm{variable}\;\mathrm{in}\;a\;\mathrm{sample}$$
$$\bar{y}=\mathrm{mean}\;\mathrm{of}\;\mathrm{the}\;\mathrm{values}\;\mathrm{of}\;\mathrm{the}\;y-\mathrm{variable}$$

Figure 13 below shows the normalized heat map where a Pearson correlation coefficient closer to −1 translates to a highly significant decreasing effect on the specific property, a value of +1 translates to a highly significant increasing effect on the specific property, and a value near zero is interpreted as an insignificant parameter to the property of interest. There are three areas present within this heat map, the Input Blend Formulation coefficients which explain how the blends are not correlated with each other, the Output Material Property coefficients which dictate how each output property is interrelated with one another, and the Input Output Correlation coefficients that give insight into how additives influence the output viscoelastic properties. To mention a few, it can be seen that $n_{\mathrm{relax}}$ is negatively correlated with tan δ and that $\sigma_{\mathrm{relax}}$ is positively correlated with hardness of the material, captured by the durometer. It can be seen within the heat map that voids have a positive Pearson correlation coefficient value with tan δ, therefore it is expected to see the rise of tan δ with increased amounts of voids.

The positive correlation of void content on tan δ specified by the Pearson correlation coefficient is confirmed in Figure 14 below where blends 8 and 9 clearly show an increasing trend for tan δ as void content increases. The above-mentioned trend is present in all blends, allowing for the prediction of tan δ behavior for samples with zero percent void content to 40% void content by extrapolating data via linear regression techniques. 

As seen in Figure 11 above, the quantity of voids in the sample influences the stresses reached during relaxation testing, an indication of rigidity. This influence on rigidity is also captured in hardness measurements, a static method in which it can be seen that increasing void content results in a decrease in hardness. Figure 15 below shows how hardness of blend 5 decreases in a linear fashion as void content increases. 

#### 3.1.2. Influence of Sulfur on Viscoelastic and Static Properties

An increased amount of sulfur within formulations results in an increased number of cross-links, and the rise in crosslink density results in a decreased ability for molecular chains to move past one another [31,60]. If mobility is limited, then intermolecular forces will be present, restricting full relaxation, and the lag between the strain and stress response will be minimized [31,32]. All three relaxation curves in Figure 16 below represent blends that only differ in the quantity of sulfur within the formulation. With no plasticizer present within the formulation, it can be appreciated how increasing the sulfur amount in fact increases rigidity, as seen in (a) where the blend with 4.0 pph of sulfur reaches a maximum stress 1.6 times larger than the blend with 0.5 pph of sulfur. Moreover, observing the normalized relaxation curves in (b) proves how decreasing molecular mobility creates a material that experiences less relaxation decay behavior. A decrease in relaxation decay behavior, a consequence of increased crosslink density, indicates that such material will take much longer to reach 1% of the maximum stress, to a material with lesser sulfur quantity.

The blends mentioned in Figure 16 above did not contain plasticizer within the formulation and comparing blends with an equal amount of plasticizer with varying amounts of sulfur has the ability of creating a more complex material. It can be seen in Figure 17 below that increasing the sulfur load, for a blend with 80 pph of plasticizer, results in larger maximum stress reached during testing, but opposite to what was observed in Figure 16, the normalized relaxation curves do not follow the same trend. It is visible that increasing the sulfur amount from 1.5 pph to 2.5 pph, all with 80 pph of plasticizer has minimal influence on the rate at which relaxation occurs and a substantial effect on the rigidity of the material.

With the Pearson correlation coefficient for sulfur on tanδ being −0.8, it is expected to see a decreasing trend on tanδ with increasing sulfur content. The expected trend can be appreciated in Figure 18 below where each blend has equal amounts of paraffin oil content but with varying sulfur content. A positive vertical shift in the curve can be appreciated as sulfur content decreases since it is known that heavily crosslinked polymers exhibit a decrease in molecular mobility [31,61].

Additionally, it can be seen from Figure 19 below that sulfur content also has an increasing influence on hardness where both blends show a linear trend with similar slopes. It is expected that the combination of additives can create a polymeric material that is extremely difficult to predict its mechanical properties. One additive may be in charge of influencing the relaxation behavior in one manner, while the second additive may counteract that influence but improve another material property. For that reason, formulating is a sensitive balance between many additives, a task that needs to consider additive-to-additive effects, thus the reason for implementing predictive tools such as linear regression, the Response Surface Method, Artificial Neural Networks, and Gaussian Process Regression for viscoelastic property predictions. Although linear regression lacks the complexity, it is able to serve as a justification for why a specific application requires a more complex method to increase accuracy. In this study, linear regression served as a baseline study to prove that the formulation process is a highly non-linear process that requires a complex system for prediction.

#### 3.1.3. Influence of Paraffin Oil on Viscoelastic and Static Properties

Paraffin Oil is commonly used in the rubber industry to decrease the blend viscosity and to facilitate processing since the ultra-high molecular weight of natural rubber is extremely viscous [43,62]. Introducing paraffin oil into the blend evidently reduces the strength of the material, which can be seen in Figure 20 below where an increase in charge of such plasticizer reduces the stresses experienced during relaxation testing at 30% strain. At the given sulfur amount of 1.5 pph it is visible how the addition of paraffin oil creates a balancing effect to the stresses reached during testing. Unfortunately, this observation is not visible when the amount of sulfur is further increased to 2.5 pph of sulfur, the sulfur amount reaches a point in which it masks the effects of paraffin oil and limits the influence on rigidity. 

As mentioned above, each additive has a limiting effect on the mechanical behavior, therefore the saturation of an additive within a blend may cause minimal influence on the system after a certain threshold is reached. This relationship can be seen in Figure 21 below where the relaxation tests which differ only in paraffin oil content show that a blend with 40 pph of paraffin oil results in a material with similar relaxation behavior to a blend with 80 pph of paraffin oil.

Similar to Figure 11 above, the max stress reached during relaxation testing can be graphed with respect to void content and paraffin oil content. Figure 22 below shows the max stresses experienced during relaxation testing for blends with 2.5 pph of sulfur while Figure 23 shows it for a blend with 1.5 pph of sulfur. Sulfur and paraffin oil have competing effects on the rigidity of the material, and it can be noted by tabulating the slopes of stress decay for each set of data points that slopes differ in trend depending on the amount of sulfur content. 

Upon evaluation of the tabulated slopes from Figure 22 and Figure 23, it is evident that the influence of voids in a 2.5 pph blend is more aggressive and becomes more pronounced as the paraffin oil content increases. For the same given paraffin oil content of 80 pph, a system with 1.5 pph of sulfur experiences a stress loss of 2.1 kPa for every percent void present in the sample while a blend with 2.5 pph of sulfur experiences a 4.4 kPa decrease in max stress during relaxation for every percent void present in the sample. As mentioned, the formulation process is a sensitive balance between many competing additives and a formulator may arrive to a similar material with several blends. Figure 24 below shows how at approximately 16% void content a material with 2.5 pph of sulfur and 1.5 pph of sulfur, all with 80 pph of paraffin oil content will have a similar rigidity in a relaxation test at 30% compressive strain.

The DMA allowed for the characterization of tanδ where it can be seen for blend 3–5, in Figure 25a below, how paraffin oil has a minimally significant increasing influence on tanδ until a threshold is reached. Upon reaching a set threshold of paraffin oil content, the trend is turned into a decreasing relationship. Once the amount of sulfur is increased from 1.5 pph to 2.5 pph, it can be seen in Figure 25b below that the increased sulfur amount overpowers the influence of paraffin oil, and the influence imparted by the plasticizer has an increasing effect on tanδ.

Although the increasing and decreasing trend of paraffin oil on tanδ are visible, it is important to note that paraffin oil, compared to the other additives has a minimal influence on tanδ. As seen in Figure 13 above, the heat map clearly shows that paraffin oil content is rated with a value of −0.093 regarding the level of significance, compared to sulfur which is rated at a value of −0.8. As mentioned before, as the values reach −1 or 1, the level of significance is greater than a value near zero. Moreover, Paraffin Oil had a decreasing effect on the hardness of the material as seen in Figure 26 below where each shade of gray specifies a different amount of paraffin oil within the blend. 

### 3.2. Predictive Methodologies for Reverse Engineering

#### 3.2.1. Response Surface Method Prediction Results

Each output response has its respective model equation, as seen in Table 4 below, where each coefficient describes the level of influence that each linear, quadratic, and interaction parameter has on the performance of the system. With hardness and $\sigma_{\mathrm{relax}}$ during relaxation being closely interrelated to the rigidity of the material, a transient property, the models for both responses reach high prediction accuracies compared to the other two viscoelastic properties where time-dependent behavior is characterized. The RSM model created from only 11 blends results in an average Prediction R^2^ of 0.89, and by comparing the proximity of each R^2^ within each respective model, it is evident that overfitting is not occurring in this model given that there is no large variation between all three R^2^ values. 

Standardizing the coefficients allows for the configuration of the Pareto Chart of the Standardized Effects where Figure 27, Figure 28, Figure 29 and Figure 30 show the Pareto Chart for each respective output response within this study. A, B, and C all represent a controllable variable (factors), as seen in the legend on the right of each Pareto Chart. A linear term within the chart only includes one factor, such as A, representing the linear term of Sulfur. A quadratic term includes two factors, such as AA, and an interaction term within the chart is shown with two different factors, such as AB, which describe the interaction between Sulfur and Paraffin Oil. 

The Pareto chart for $\sigma_{\mathrm{relax}}$ and hardness both show that Paraffin Oil has the largest influence on the output response and that the response should closely follow a linear trend since the largest terms only have one factor. This linear relationship can be confirmed by looking at Figure 22 above where the linear trendline is present with an R^2^ of 0.99. Observing the Pareto charts for $n_{\mathrm{relax}}$ and tanδ both show how complex the model is as the top two most influential terms are quadratic terms. RSM suggests that if the coefficient of the squared term is significant, then one can conclude that the relationship between both controllable variables closely follows a curved response. This statement is supported in Figure 31 where the relationship of sulfur on $n_{\mathrm{relax}}$ does not follow a linear trend, but more of an asymptotic curve.

#### 3.2.2. Artificial Neural Network Prediction Results

The final ANN architectures selected are summarized in Table 5, and the parity plots of the experimental and predicted values from 5-fold cross-validation are illustrated in Figure 32. For each of the four material properties, the $R^{2}$ value increased compared to the linear regression baseline. It is worth noting that a simple NN architecture has the ability to capture material-property correlation pertinent to the rubber blend content, especially for $n_{\mathrm{relax}}$ and tanδ whose underlying functions are more nonlinear, as suggested by the significant improvements of the two. When compared with RSM, ANNs also showed improvements in prediction accuracies for $n_{\mathrm{relax}}$ and tanδ. This again confirms that $n_{\mathrm{relax}}$ and tanδ would benefit from a more nonlinear modeling approach. Although the selected ANN architecture for $\sigma_{\mathrm{relax}}$ performed slightly worse than RSM, some of the more complex architectures (with over 1000 parameters) tested could reach a comparable $R^{2}$ value. However, to avoid overtraining, the model complexity and accuracies were leveraged during the model selection process. 

After the models were developed, the sensitivity analysis was performed on both the linear regression baseline and the ANNs. For linear regressions, the gradients were simply the weights, and for ANNs, the gradients were obtained from backpropagation. Figure 33 shows the averaged gradients from the individual folds of cross-validation. In general, the gradients from both the LR and ANN models have the same trend except for $n_{\mathrm{relax}}$, where the paraffin oil content has a positive gradient in LR but negative gradient in ANN. The RSM analysis also suggests a negative gradient, which corroborates with relaxation theory as the plasticizer increases the free volume between the rubber molecules, further facilitating relaxation as you increase plasticizer content.

#### 3.2.3. Gaussian Process Regression Prediction Results

The parity plots for GPR are illustrated in Figure 34, with the shaded region suggesting the bounds for 95% confidence intervals (obtained by 1.96 standard deviations given by GPR prediction). For each of the four material properties, the $R^{2}$ value increased significantly compared to all the above methods, and the uncertainties are small in regions where the experimental data are abundant. In spite of the ability to make predictions with uncertainty, it is difficult to obtain physical insights from GPR due to its nonparametric nature. However, given the high cross-validation accuracy, GPR could potentially be used to generate synthetic data for the Bayesian optimization algorithm that aims to find the optimal blend content. 

### 3.3. Material Design Optimization

With industries, such as the footwear industry moving towards sustainable material alternatives for midsoles, target properties could be selected based on an existing footwear material via characterization techniques. The respective methods would be implemented to predict the blends that would theoretically exhibit the target properties. In this section, we show that the previously developed predictive models can be used to aid reverse engineering.

RSM provides an empirical model that allows users to utilize it as a method for reverse engineering based on target parameters. Similarly, the trained ANN and GPR could also be used for reverse engineering. Based on the target parameters, a score function was constructed based on the weighted Euclidean distance between the predicted and the target values. The optimal set of parameters is the ones that minimize the score function. The input rubber blend compositions are constrained by experimental considerations, such as upper and lower bounds that are physically meaningful and precision limits of the characterization equipment. Table 6 below shows the blends outputted by each predictive method which exhibit the target properties mentioned within it. 

For ANN and GPR, since the input design space is relatively small, we generated all the possible input rubber blend compositions in the input design space within the feasible region. This is possible due to the consideration of experimental precision that makes the distribution of the design space rather discrete. Overall, around 6600 possible blend compositions and their corresponding predicted properties were obtained. The composition with the minimum score (loss) was identified as the optimal blend composition.

The above reverse engineering method is relatively straightforward and fast to populate given this small design space. In fact, all the predicted values were generated within seconds. Nevertheless, when we have a large design space or when the input compositions are no longer discrete, this method may fail. Alternatively, we used Bayesian optimization (BO) [63] to find the optimal blend compositions. BO utilizes Gaussian processes (as in GPR) to solve the black-box optimization problem—in this case, a minimization problem with an objective function to minimize the score while satisfying the design parameter constraints [64]. The algorithm first establishes a surrogate model that computes a posterior distribution (mean and variance) of the objective function using a set of sample points. The surrogate model is then used to construct an acquisition function that estimates the distribution of the objective function for any test data. A recommended blend composition can be obtained by minimizing the acquisition function (where we used the score function); this blend composition is used to obtain a new observation data point (usually from an experiment or from some existing predictive model) to be incorporated for updating the surrogate model. After running these steps iteratively, the algorithm will converge to an optimal blend composition. 

In this study, instead of collecting data from experiments for each iteration, we used the previously trained GPR model to generate the new “observations” given its high CV accuracy. The BO was implemented using Scikit-learn [51] (version 0.24.2) and Scipy [65] (version 1.7.0). The resulting blend composition from BO is similar to the findings from the previous approach. Overall, even though BO bypasses the construction of specific predictive models for material properties, it can guide the design of new experiments and thus reduce the overall experimental costs.

## 4. Conclusions

The design process may include various iterative cycles, a time-consuming task for one with limited experience within the field, and with industry aiming to maximize productivity, each iterative cycle in the design process equates to a loss of resources. The presented methods in this research study provide users with low-fidelity and high-fidelity design optimization methods applicable for reverse engineering applications. These methods provide users with an automated formulation process capable of informing the user of key material properties that each formulation will have. Table 7 below briefly highlights each method where accuracy grows as you move upwards in algorithm development complexity.

By selecting the appropriate viscoelastic properties and predictive methodologies, the small number of experimental blends captured the highly non-linear behavior of natural rubber blends, which concluded with GPR showcasing the largest 5-fold cross-validation predictive accuracy of 100%. GPR was able to accurately predict the short-term behavior, long-term viscoelastic behavior, transient and dynamic properties. As depicted in Table 6 above, by specifying the desired material properties of the product, each algorithm was able to output an optimal blend capable of behaving similarly to what was desired. The algorithm provides the user with a starting point based on a highly accurate algorithm, saving the user time, and ultimately saving valuable resources.

## Figures and Tables

**Figure 1 polymers-14-02262-f001:**
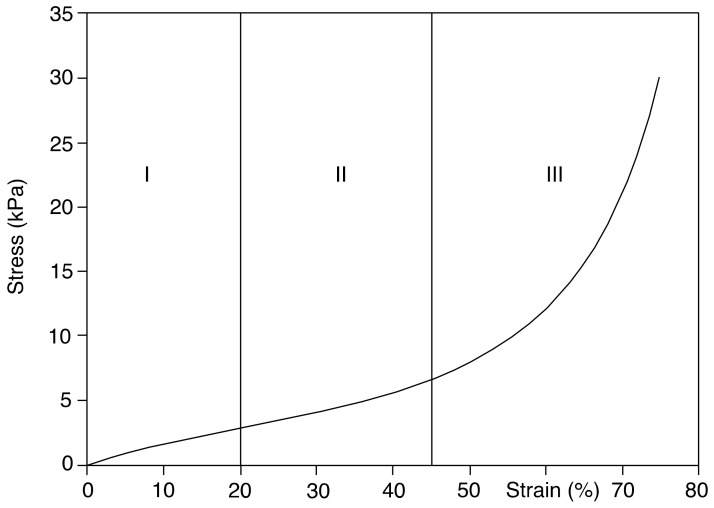
A typical compression curve for a foamed elastomeric material where (**I**) is the initial region with larger tangent modulus, (**II**) is the buckling region with the reduced tangent modulus, and (**III**) is the densification zone.

**Figure 2 polymers-14-02262-f002:**
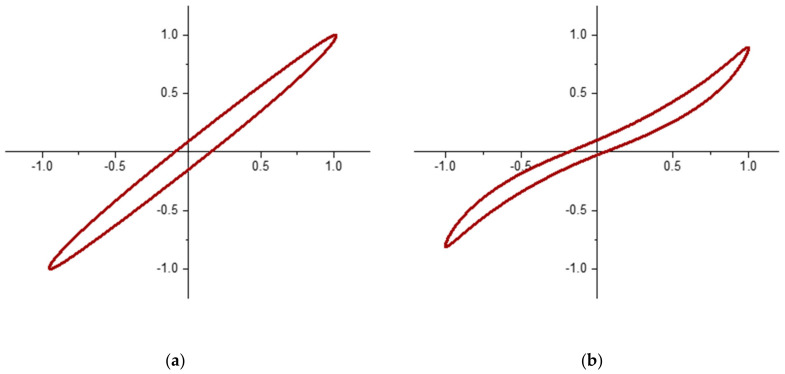
(**a**) Lissajous curve of raw natural rubber, a nearly-linearly viscoelastic material, (**b**) Lissajous curve of standard athletic footwear material, a non-linearly viscoelastic material.

**Figure 3 polymers-14-02262-f003:**
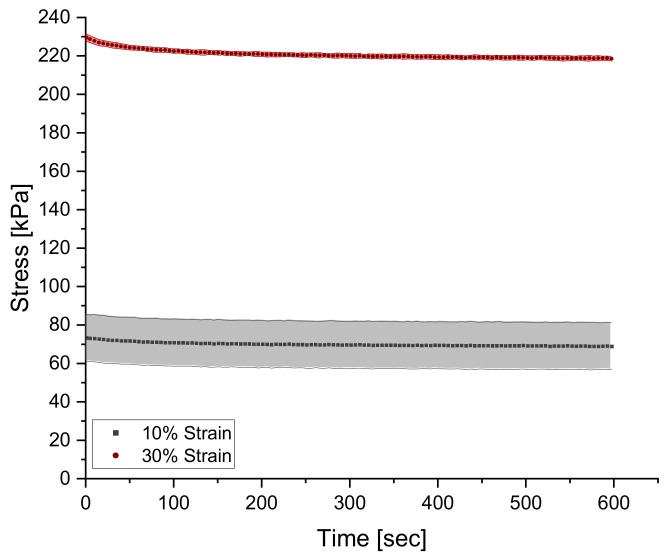
A comparison between a 10% and 30% strain test showcasing the low signal-to-noise ratio for the lower strain-level test.

**Figure 4 polymers-14-02262-f004:**
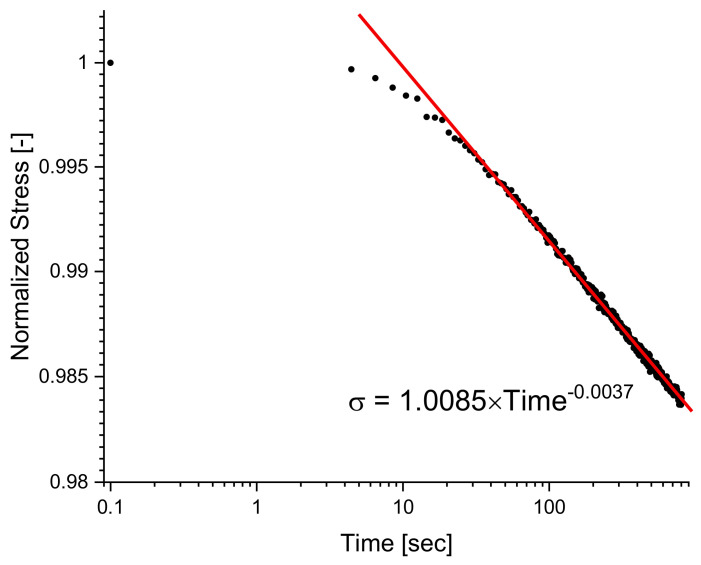
This shows the raw relaxation curve which can be fit to a power function to quantify the long-term behavior under relaxation.

**Figure 5 polymers-14-02262-f005:**
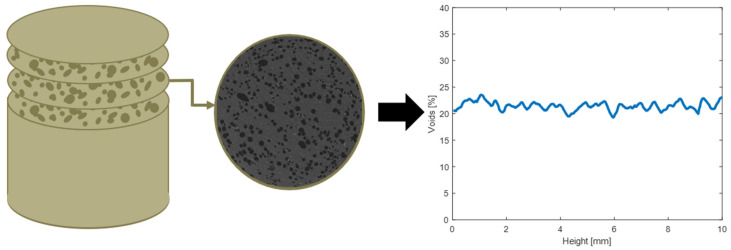
Depicts the workflow for the MATLAB program for quantifying the quantity of voids present within the sample.

**Figure 6 polymers-14-02262-f006:**
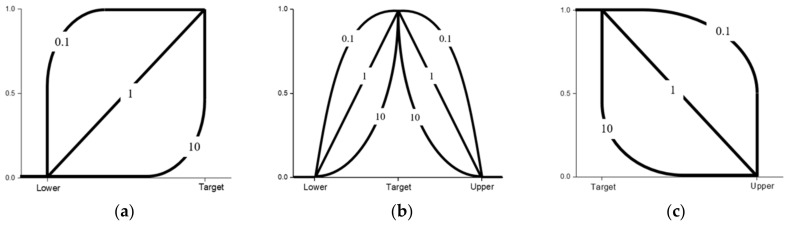
Desirability functions for different goals and how weights influence their respective shapes. (**a**) Minimize the response, (**b**) Achieve target value, and (**c**) Maximize the response.

**Figure 7 polymers-14-02262-f007:**
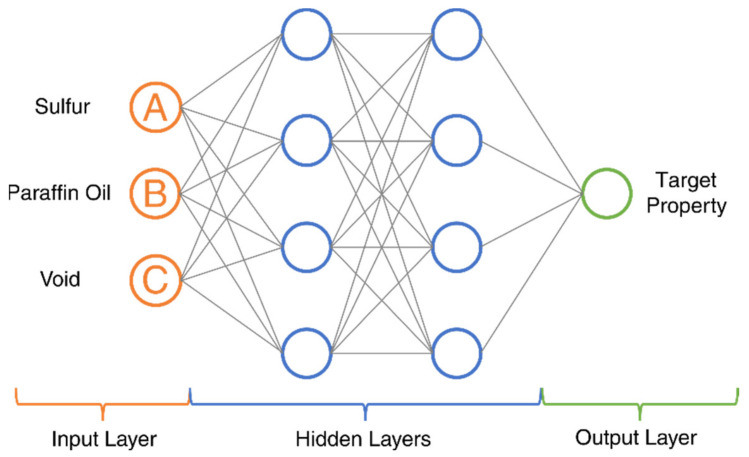
ANN basic architecture.

**Figure 8 polymers-14-02262-f008:**
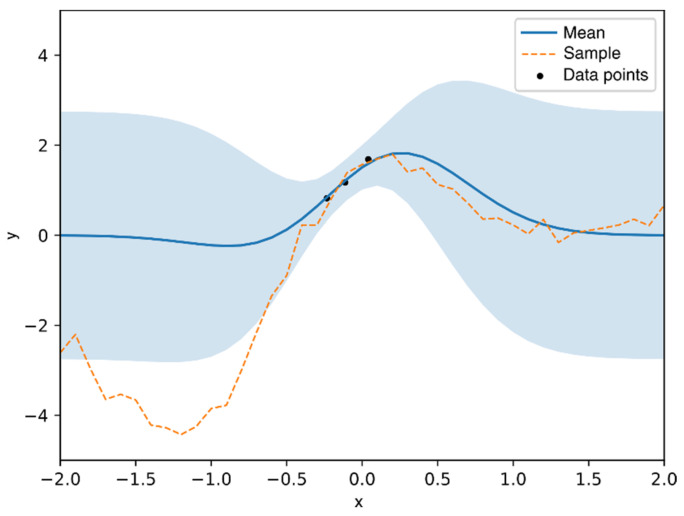
An illustration of GPR and how more data increases the predictive capabilities.

**Figure 9 polymers-14-02262-f009:**
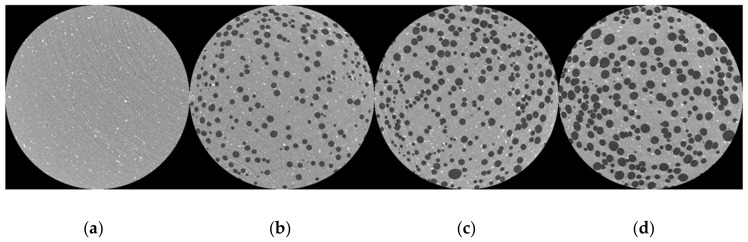
(**a**) μCT scan of a sample with 0% voids. (**b**) μCT scan of sample with 11.8% voids. (**c**) μCT scan of sample with 19% voids. (**d**) μCT scan of sample with 32.2% voids.

**Figure 10 polymers-14-02262-f010:**
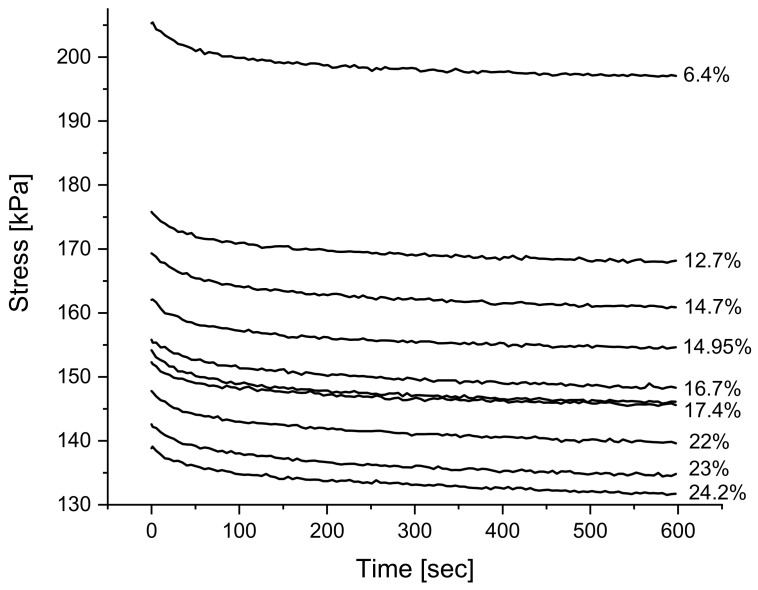
An overlay of relaxation tests of blend 1 at varying levels of void content mentioned in the text to the right of each curve.

**Figure 11 polymers-14-02262-f011:**
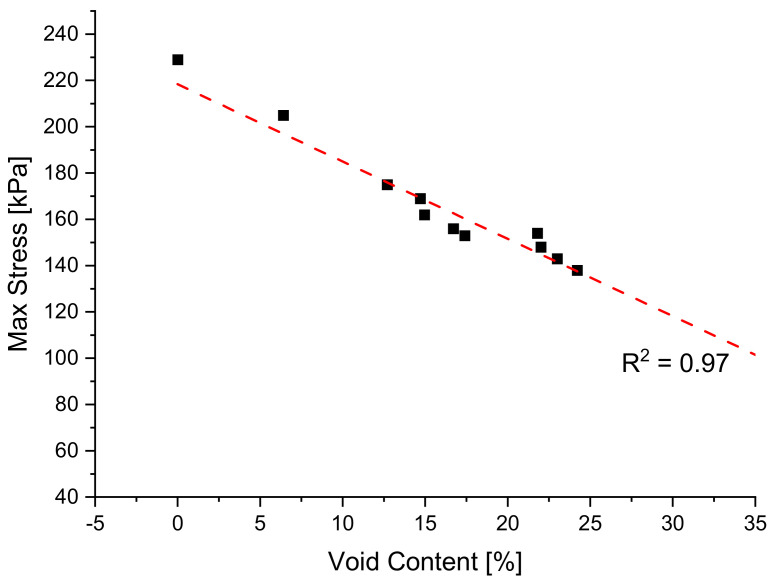
Depiction of the linear relationship between max stress experienced in relaxation testing and void content for blend 1.

**Figure 12 polymers-14-02262-f012:**
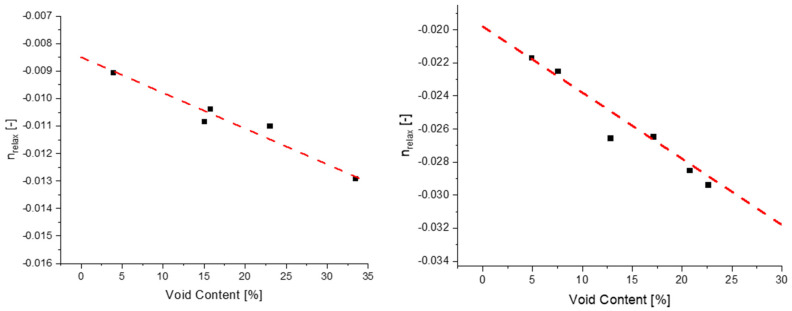
The relationship between voids and the rate at which stress decays for blend 9 (**left**) and 10 (**right**), characterized by $n_{\mathrm{relax}}$.

**Figure 13 polymers-14-02262-f013:**
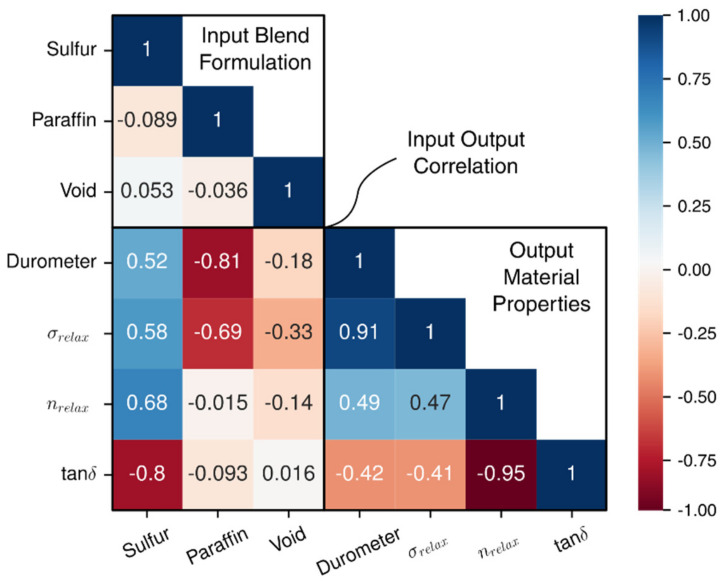
The Pearson correlation coefficients for each parameter.

**Figure 14 polymers-14-02262-f014:**
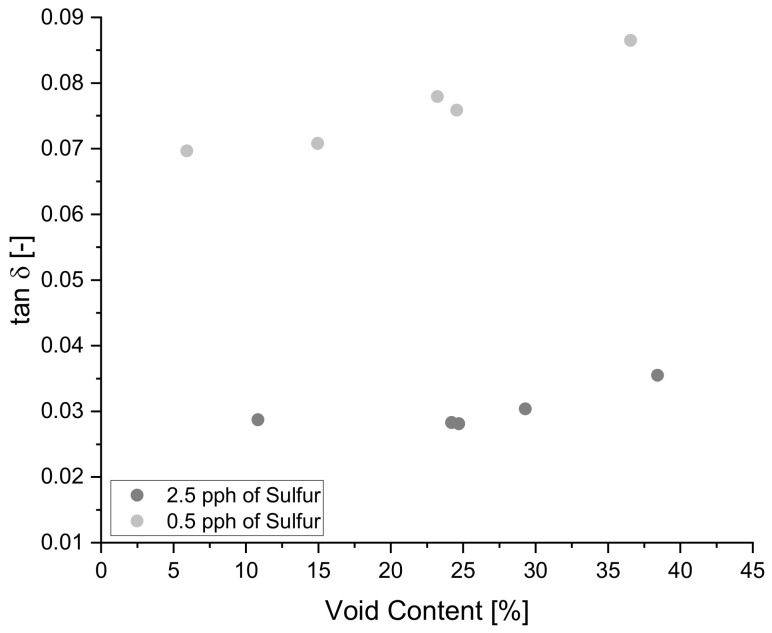
The increasing relationship of void content on tan δ.

**Figure 15 polymers-14-02262-f015:**
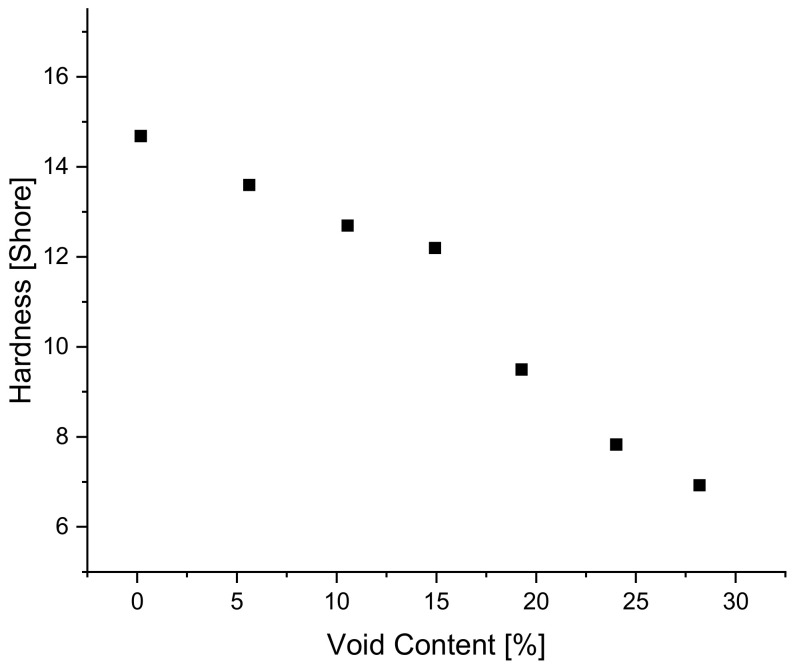
The influence of void content on hardness.

**Figure 16 polymers-14-02262-f016:**
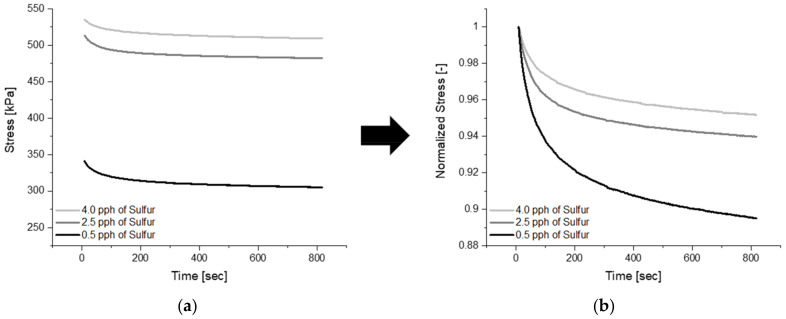
(**a**) Depicts relaxation curves for blends 7, 8 and 9 while (**b**) represents the normalized curves.

**Figure 17 polymers-14-02262-f017:**
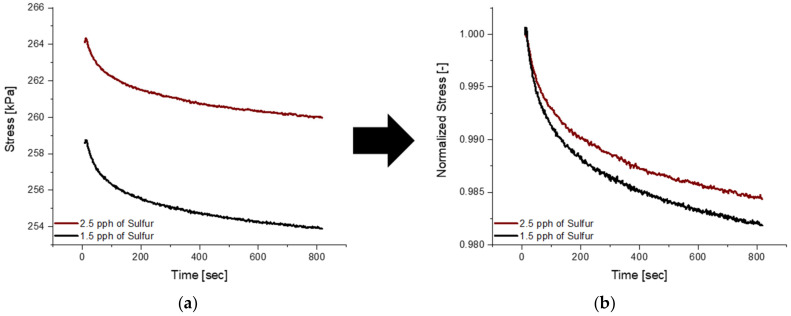
(**a**) Depicts relaxation curves for blends 1 and 4 while (**b**) represents the normalized curve, showing a large similarity in regard to the stress decay behavior.

**Figure 18 polymers-14-02262-f018:**
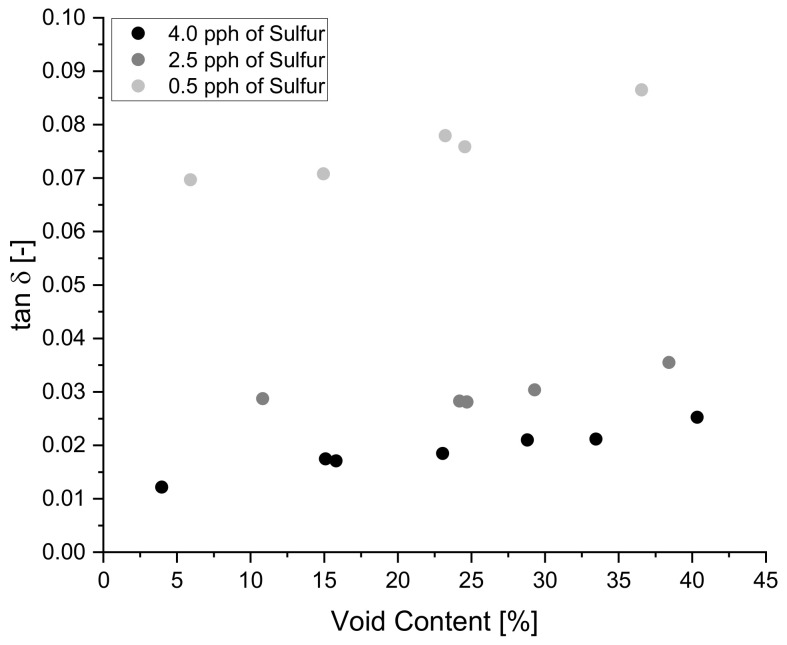
The influence of sulfur content and void content on tan δ for blends 7, 8 and 9.

**Figure 19 polymers-14-02262-f019:**
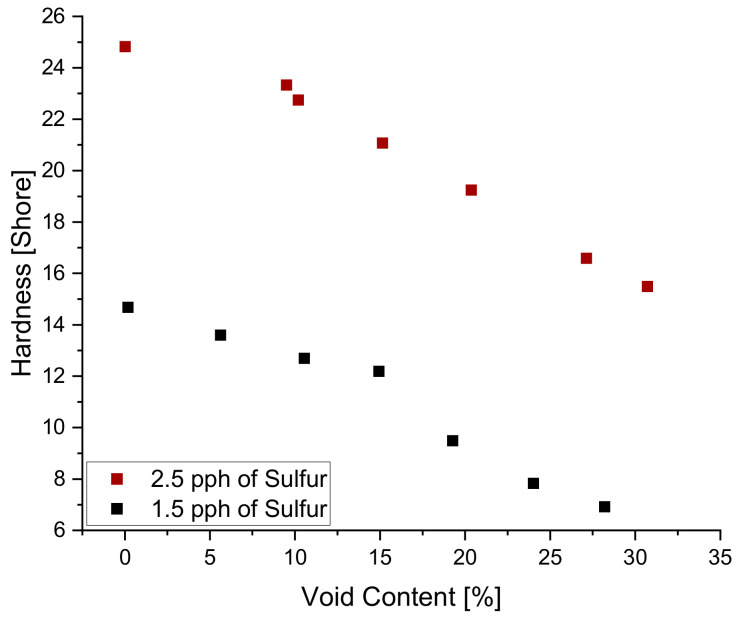
The influence of sulfur content and void content on hardness.

**Figure 20 polymers-14-02262-f020:**
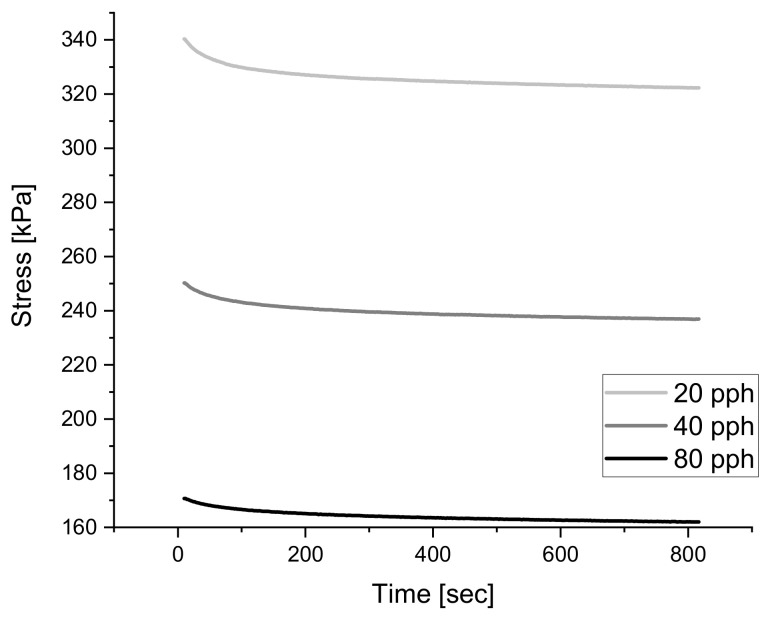
The influence of paraffin oil content on relaxation behavior of NR blend with 1.5 pph of sulfur.

**Figure 21 polymers-14-02262-f021:**
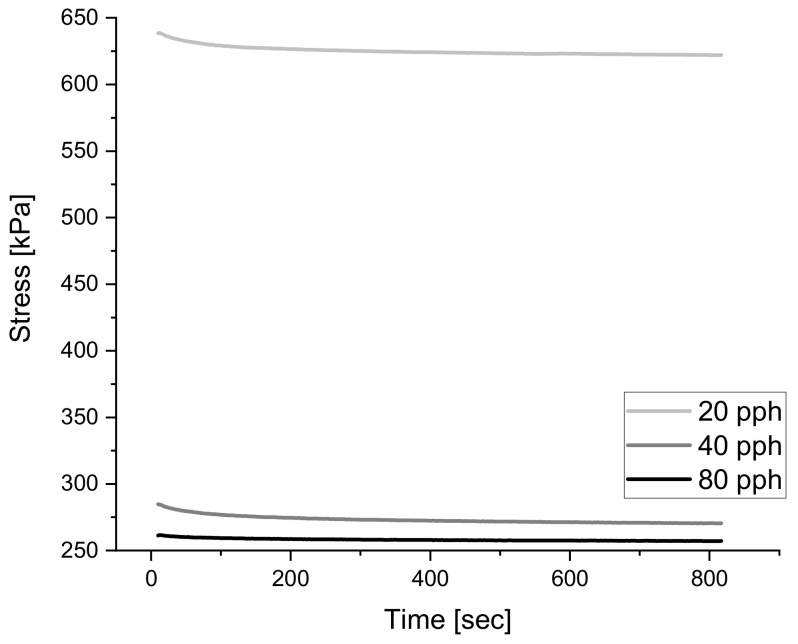
The influence of paraffin oil content on relaxation behavior of NR blend with 2.5 pph of sulfur.

**Figure 22 polymers-14-02262-f022:**
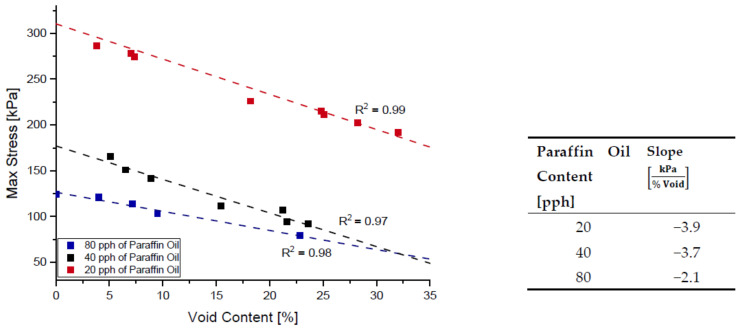
The influence of paraffin oil content on $\sigma_{\mathrm{relax}}$ of NR blend with 1.5 pph of sulfur.

**Figure 23 polymers-14-02262-f023:**
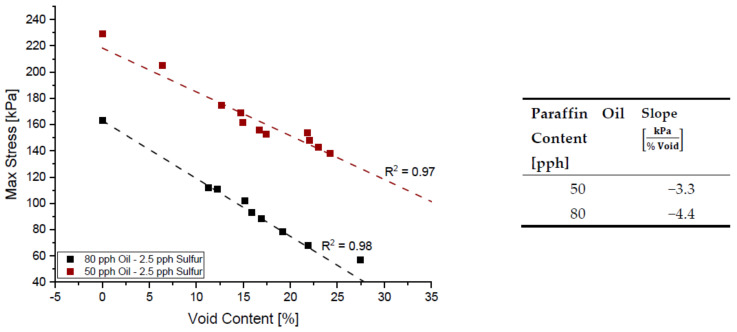
The influence of paraffin oil content on $\sigma_{\mathrm{relax}}$ of NR blend with 2.5 pph of sulfur.

**Figure 24 polymers-14-02262-f024:**
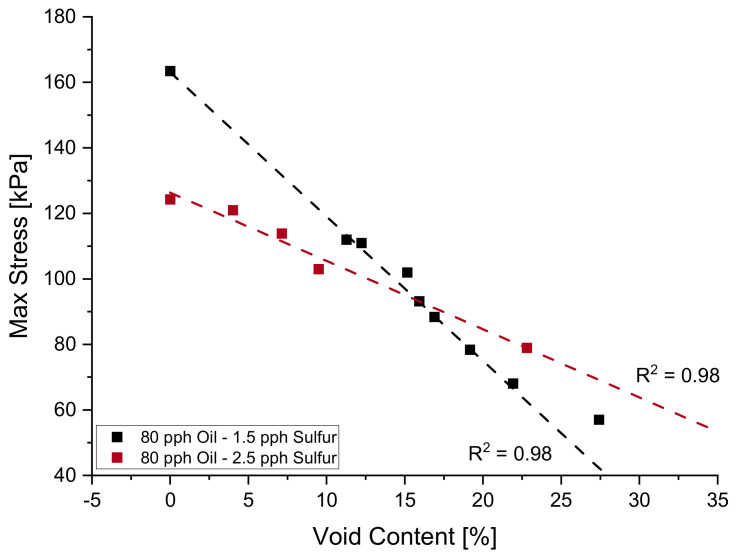
Plot describing the similarity of unique blends by varying void content.

**Figure 25 polymers-14-02262-f025:**
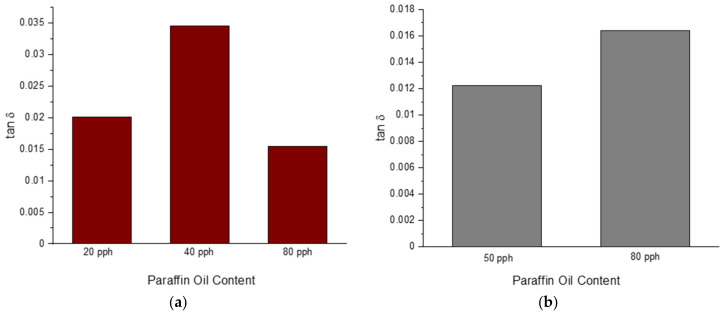
The influence of paraffin oil content on tanδ for a blend with 1.5 pph (**a**) and 2.5 pph of sulfur (**b**).

**Figure 26 polymers-14-02262-f026:**
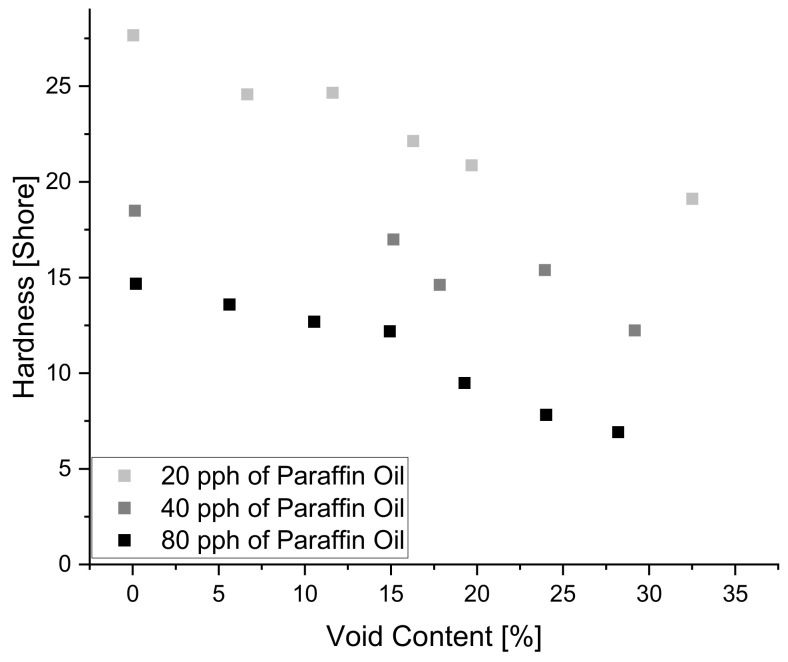
The influence of paraffin oil content on hardness.

**Figure 27 polymers-14-02262-f027:**
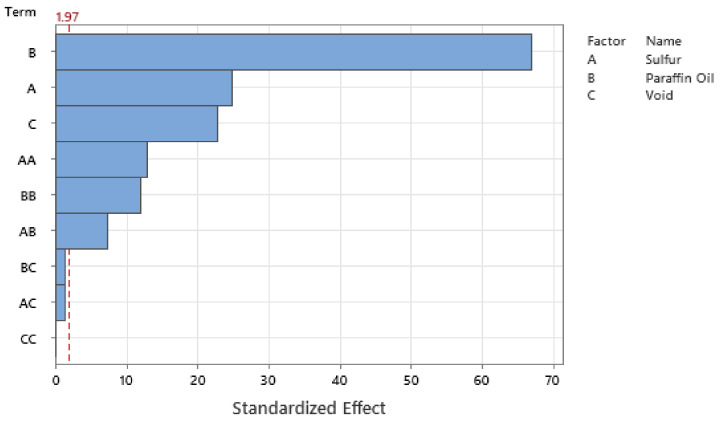
The Pareto Chart of Standardized Effects for durometer reading.

**Figure 28 polymers-14-02262-f028:**
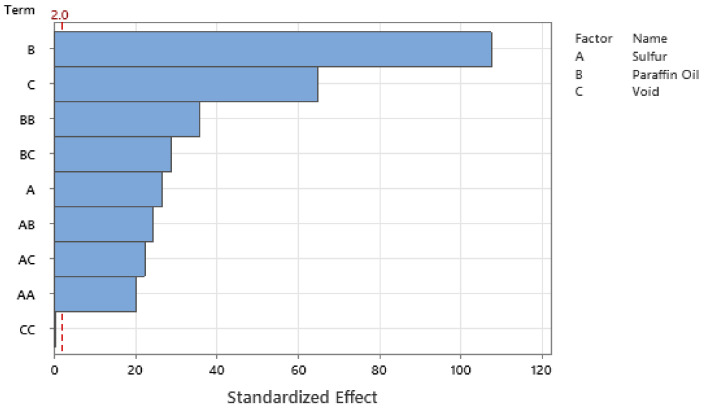
The Pareto Chart of Standardized Effects for $\sigma_{\mathrm{relax}}$.

**Figure 29 polymers-14-02262-f029:**
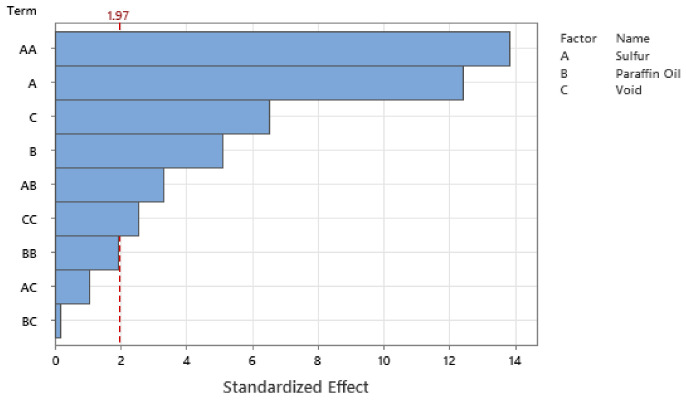
The Pareto Chart of Standardized Effects for $n_{\mathrm{relax}}$.

**Figure 30 polymers-14-02262-f030:**
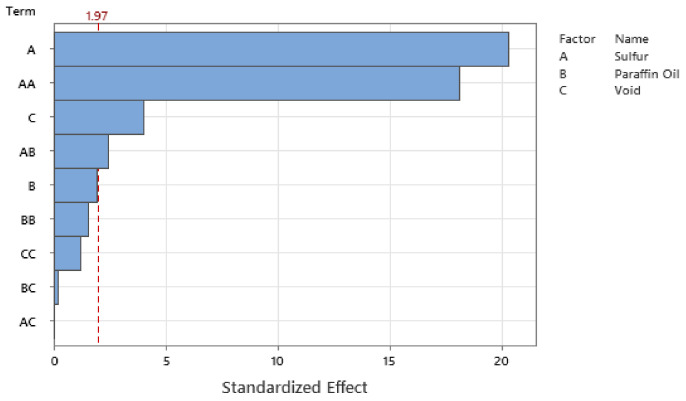
The Pareto Chart of Standardized Effects for tanδ.

**Figure 31 polymers-14-02262-f031:**
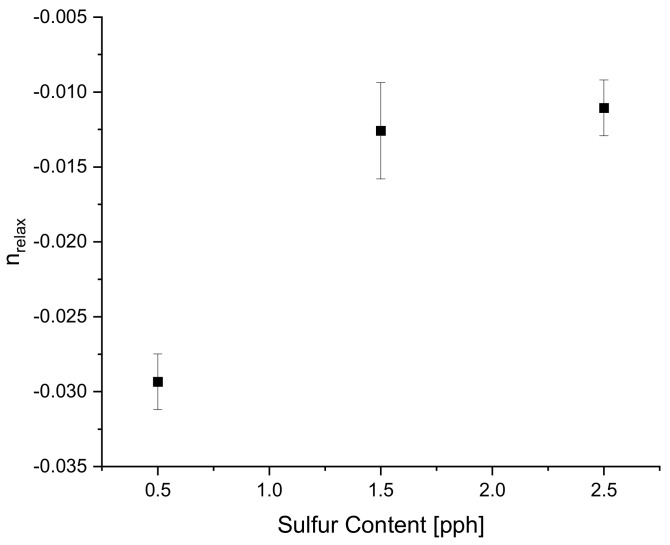
The curved response of sulfur content on $n_{\mathrm{relax}}$, further confirming the results in the Pareto chart.

**Figure 32 polymers-14-02262-f032:**
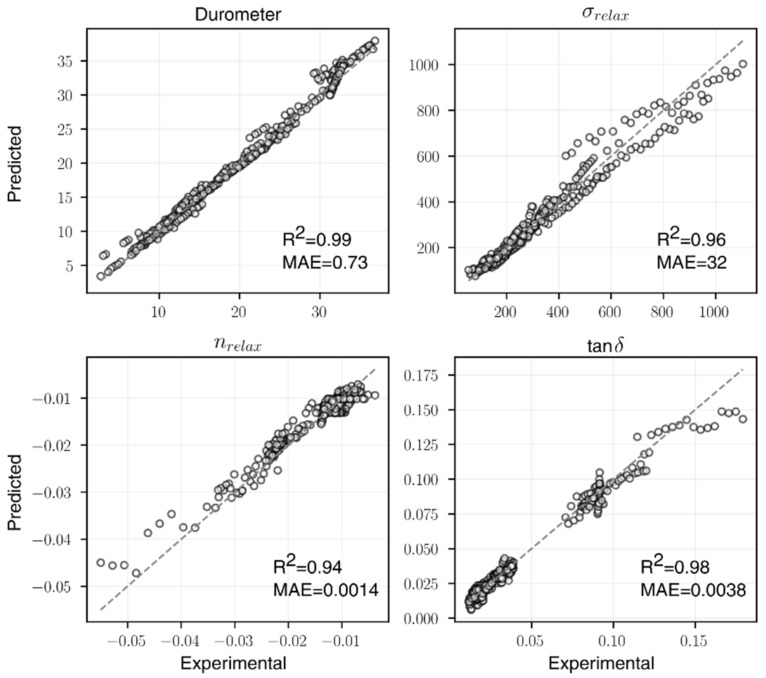
The parity plots for all four ANN models.

**Figure 33 polymers-14-02262-f033:**
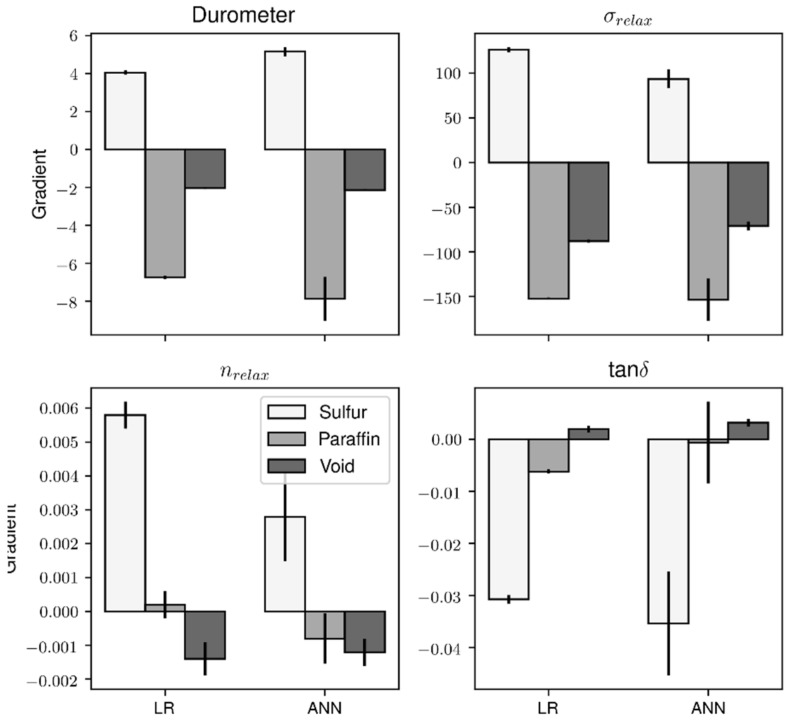
The results from the sensitivity analysis for both the linear regression baseline and the ANNs.

**Figure 34 polymers-14-02262-f034:**
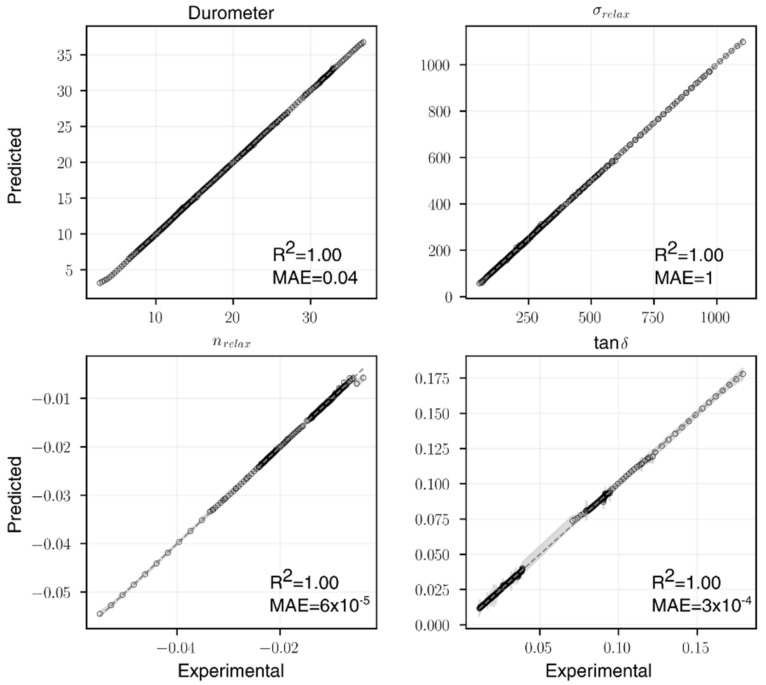
The parity plots describing Predicted vs. Experimental for GPR.

**Table 1 polymers-14-02262-t001:** Below depicts the materials used in this study.

Material	Provider	Purpose
Raw Natural Rubber	SOAN Laboratories	Raw Material
Sulfur	Fisher Scientific	Cross-linking Agent
Sodium Bicarbonate	Sigma-Aldrich	Foaming Agent
Stearic Acid	Fisher Scientific	Activator
Paraffin Oil	Fisher Scientific	Plasticizer
TMTD	Fisher Scientific	Accelerator
Zinc Oxide	Fisher Scientific	Accelerator

**Table 2 polymers-14-02262-t002:** Below depicts each blend formulation.

Blend No.	Sulfur (pph)	Paraffin Oil (pph)	Sodium Bicarbonate (pph)
1	2.5	80	8
2	2.5	50	8
3	1.5	40	8
4	1.5	80	8
5	1.5	20	8
6	0.5	20	8
7	0.5	0	8
8	2.5	0	8
9	4	0	8
10 *	0.6	39	8
11 *	0.3	59	8

* These are blends outputted by the ANN algorithm to improve accuracy.

**Table 3 polymers-14-02262-t003:** µCT scanning parameters.

Variable	Symbol
Voltage [kV]	75
Current [μA]	100
Integration Time [ms]	1000
Gain [-]	8
Number of Projections [-]	1000
Resolution [μm]	4

**Table 4 polymers-14-02262-t004:** Below depicts each blend formulation.

Response Variable	Model Equation	$R^{2}$	$Adjusted\;R^{2}$	$Predicted\;R^{2}$
Durometer [Shore A]	$25.76+7.14A-0.44B-0.26C-1.15A^{2}+0.002B^{2}+0.0001C^{2}\;+0.03\mathrm{AB}+0.01\mathrm{AC}-0.0005\mathrm{BC}$	0.9689	0.9680	0.9667
$\sigma_{\mathrm{relax}}$ [kPa]	$440.12+268.26A-10.80B-9.90C-26.37A^{2}+0.07B^{2}\;+0.01C^{2}-1.39\mathrm{AB}-2.94\mathrm{AC}+0.14\mathrm{BC}$	0.9906	0.9903	0.9900
$n_{\mathrm{relax}}$ [-]	$-0.024+0.01674A-0.00018B-0.00049C-0.003A^{2}\;+0.000001B^{2}+0.000008C^{2}+0.0000034\mathrm{AB}\;+0.000025\mathrm{AC}+0.000000\mathrm{BC}$	0.7533	0.7463	0.7367
tanδ [-]	$0.119-0.0819A+0.0004B+0.00079C+0.014A^{2}\;-0.000002B^{2}-0.000013C^{2}-0.000083\mathrm{AB}\;-0.000003\mathrm{AC}-0.000001\mathrm{BC}$	0.8748	0.8712	0.8660

**Table 5 polymers-14-02262-t005:** Below shows the predictive capabilities of the ANN.

Target Material Property	Number of Hidden Layers	Number of Neurons in the Hidden Layer	Learning Rate	Number of Learnable Parameters in the Model	5-Fold CV $R^{2}\;\mathrm{of}\;\mathrm{ANN}$	5-Fold CV $R^{2}\;\mathrm{of}\;\mathrm{Linear}\;\mathrm{Regression}\;\mathrm{Baseline}$
**Durometer**	1	64	0.01	321	0.99	0.91
$\sigma_{\mathrm{relax}}$	2	16.16	0.01	353	0.96	0.90
$n_{\mathrm{relax}}$	2	16.16	0.01	353	0.94	0.48
tanδ	1	32	0.003	161	0.98	0.67

**Table 6 polymers-14-02262-t006:** Below summarizes the blends that were classified as the optimal blend based on target properties.

Modeling Method	Prediction Results
**Target Properties**	$\sigma_{\mathrm{relax}}$: 90 $n_{\mathrm{relax}}$: −0.0527 tanδ: 0.066
RSM	Sulfur: 0.65 Paraffin: 69.5 Void: 30.0
RSM-predicted Properties	Durometer: 0.45 $\sigma_{\mathrm{relax}}$: 90.26 $n_{\mathrm{relax}}$: −0.02875 tanδ: 0.0982
ANN	Sulfur: 0.55 Paraffin: 54 Void: 17
ANN-predicted Properties	Durometer: 5.612 $\sigma_{\mathrm{relax}}$: 82.45 $n_{\mathrm{relax}}$: −0.0399 tanδ: 0.113
GPR	Sulfur: 0.55 Paraffin: 57 Void: 10
GPR-predicted Properties	Durometer: 6.187 $\sigma_{\mathrm{relax}}$: 111.78 $n_{\mathrm{relax}}$: −0.0347 tanδ: 0.114

**Table 7 polymers-14-02262-t007:** Summary of algorithm accuracies presented in this paper ranging in complexity.

Target Material Property	$\mathrm{LR}\text{—}\mathrm{Baseline}\;R^{2}$	$\mathrm{RSM}\;R^{2}$	$\mathrm{Optimized}\;\mathrm{ANN}\;R^{2}$	$\mathrm{Optimized}\;\mathrm{GPR}\;R^{2}$
**Durometer**	0.91	0.97	0.99	1.00
** $\sigma_{\mathrm{relax}}$ **	0.90	0.99	0.96	1.00
** $n_{\mathrm{relax}}$ **	0.48	0.75	0.94	1.00
** tanδ **	0.67	0.87	0.98	1.00

## Data Availability

Not applicable.

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
