# Peer review of "Natural Rubber Blend Optimization via Data-Driven Modeling: The Implementation for Reverse Engineering"

_polymers, 2022, doi:10.3390/polym14112262_

Round 1
Reviewer 1 Report
The authors presented an approach for optimizing the natural rubber blend optimization by using Data-driven modeling. In terms of content and volume, the study is scientifically sound.
The manuscript is well written however there are a few concerns.
At the end of the introduction, there is no information on what will be done in this work and what the authors want to contribute to this article?
It is unclear since the authors bring in machine learning and long-term behavior in the same paragraph and it seems like there is a missing link.
Jacob and coworkers need a citation.
pph? pounds per hour? if so specify and use SI units.
DOE--> design of experiments? please mention it the first time.
Table 2: results should be indicated in the result section and not before even presenting the ANN algorithm.
Resolution of µCT? and acquisition time.
What was the reason for using different ANN parameters? Isn't it simple to use the same parameters? Also, why didn't the authors combine all three target properties?
Also, what was the training and testing size? Was the same data set used for both?
Author Response
Overall, thank you for the very useful feedback and constructive comments. The comments have been addressed and discussed below. I look forward to hearing back.
The authors presented an approach for optimizing the natural rubber blend optimization by using Data-driven modeling. In terms of content and volume, the study is scientifically sound.
The manuscript is well written however there are a few concerns.
At the end of the introduction, there is no information on what will be done in this work and what the authors want to contribute to this article?
Thank you for the great suggestion. A paragraph was added at the end of the introduction to explain the goal of this study and how data-driven modeling techniques were used within the study.
It is unclear since the authors bring in machine learning and long-term behavior in the same paragraph and it seems like there is a missing link.
The introduction now has a paragraph clearly stating the goal of this study in regards to why machine learning was implemented.
Jacob and coworkers need a citation.
This citation has been added to the text and also the reference list has been updated accordintly. Thank you for catching that mistake.
pph? pounds per hour? if so specify and use SI units.
The abbreviation stands for parts per hundred of rubber. I have specified pph (parts per hundred of rubber) at the first instance pph is shown within the paper.
DOE--> design of experiments? please mention it the first time.
The abbreviation has been defined in the first time seen within the paper. Thank you for catching it.
Table 2: results should be indicated in the result section and not before even presenting the ANN algorithm.
Thank you for the suggestion. The algorithms have the ability of showing you where gaps might be present within your dataset. We believe including the asterisk is appropriate as the model development included those 2 additional formulations which were in charge of possibly increasing accuracy. Similar to statistical software, the software may suggest before conducing the full study, some recommended blends.
Resolution of µCT? and acquisition time.
The resolution and scanning time has been mentioned within the manuscript.
What was the reason for using different ANN parameters? Isn't it simple to use the same parameters? Also, why didn't the authors combine all three target properties?
Also, what was the training and testing size? Was the same data set used for both?
Both comments are great suggestions. As mentioned in the results, each material property has its own optimal parameters as some may be more non-linear than others. Instead of making one generic ANN, the best solution is achieved via unique models. Also, the results section specifies that it is a 5-fold cross validation technique. This means that the data is split in 80% training and 20% testing.
Reviewer 2 Report
The authors performed a study on reverse engineering rubber properties based on existing data by building models that connect three major components (sulfur content, paraffin oil content and void content) to the viscoelastic properties of rubbers. The study also discussed the detailed property of each characterization from the typical samples. Four computational models (LR, RSM, ANN, GPR) are used to build the predictive model based on the existing data set. The authors also proposed a method for performing the reverse engineering of the rubber properties. The design of the study is logically reasonable and technically sounds. The proposed model works well for these rubber samples. I would like to recommend this work to be published on Polymers after the authors address the following issues:
(1) In Page 5, section 2.2.3, the authors mentioned signal-to-noise (S/N) ratio. Here, a higher S/N ratio should mean the noise level is low compared with the value of the data points. Same issue also found in fig 3.
(2) In Page 6, at the bottom paragraph, the authors used darker color to show the voids, so the pixel density should be lower for void instead of a higher pixel density mentioned in the text.
(3) In Page 9, the authors introduced the ANN model for modeling the property and composition. The amount of training data set is 326 as mentioned in the text. However, in table 2, the authors only listed 11 data samples. Could the authors provide additional discussion about how the 326 data points come from? Here I have another question connect to this question. The ANN model is usually suffered from the lack of training data points problem in the training stages. A data augmentation is sometime needed. Is 326 data points enough to training the proposed ANN model?
(4) In Page 12, at the bottom of the paragraph, “will by…” should be “will be…”.
(5) In Page 12 and 13, Fig 12 and fig 13 should be combined side by side or at least use same scale of n_relax for better comparison.
(6) In Page 13, the authors should provide the equation for calculating the Pearson correlation coefficient for the convenience of the readers.
(7) In Page 20, fig 26, the y axis scale should be same for easy comparison.
Author Response
Thank you for the constructive and extremely useful feedback. The answers and discussions of the comments are seen below. Thank you for taking the time to read the manuscript, it is highly appreciated.
The authors performed a study on reverse engineering rubber properties based on existing data by building models that connect three major components (sulfur content, paraffin oil content and void content) to the viscoelastic properties of rubbers. The study also discussed the detailed property of each characterization from the typical samples. Four computational models (LR, RSM, ANN, GPR) are used to build the predictive model based on the existing data set. The authors also proposed a method for performing the reverse engineering of the rubber properties. The design of the study is logically reasonable and technically sounds. The proposed model works well for these rubber samples. I would like to recommend this work to be published on Polymers after the authors address the following issues:
- In Page 5, section 2.2.3, the authors mentioned signal-to-noise (S/N) ratio. Here, a higher S/N ratio should mean the noise level is low compared with the value of the data points. Same issue also found in fig 3.
Thank you for finding that mistake. The correct wording has been added.
- In Page 6, at the bottom paragraph, the authors used darker color to show the voids, so the pixel density should be lower for void instead of a higher pixel density mentioned in the text.
I have added some additional information to increase comprehension of the thresholding technique. Please let us know if additional information is needed.
- In Page 9, the authors introduced the ANN model for modeling the property and composition. The amount of training data set is 326 as mentioned in the text. However, in table 2, the authors only listed 11 data samples. Could the authors provide additional discussion about how the 326 data points come from? Here I have another question connect to this question. The ANN model is usually suffered from the lack of training data points problem in the training stages. A data augmentation is sometime needed. Is 326 data points enough to training the proposed ANN model?
Below figure 14 states that extrapolation of the other data was achieved via regression techniques. Additionally, as mentioned in the results and development section, ANN performance was monitored via cross validation, early stopping and drop out layers to ensure a suitable model was developed. ANN are able to be used in all types of data sets as long as the proper precautions are taken during the development. In this case, a high 5 fold CV accuracy model developed which was protected against overfitting.
- In Page 12, at the bottom of the paragraph, “will by…” should be “will be…”.
Thank you. It has been corrected.
- In Page 12 and 13, Fig 12 and fig 13 should be combined side by side or at least use same scale of n_relax for better comparison.
Thank you for the suggestion. The plots have been placed side-by-side but the scale was not changed given that they are very different in magnitudes which will not allow the reader to observe changes in slope.
- In Page 13, the authors should provide the equation for calculating the Pearson correlation coefficient for the convenience of the readers.
Very great suggestion to ensure readers understand how the heat map is constructed. We have added the equation with a small explanation of what each variable represents
(7) In Page 20, fig 26, the y axis scale should be same for easy comparison.
Similar to comment 5, adjusting the scale would not allow for easy comparison given that their magnitudes are very different. This adjustment would ultimately not allow the reader to fully comprehend the changes in magnitude for the specific conditions.
Round 2
Reviewer 1 Report
Authors competently replied to every remark raised!
It is good to go to the press
Reviewer 2 Report
Accept as is.